# Solar cycle signatures in lightning activity

**Jaroslav Chum**[1], **Ronald Langer**[2], **Ivana Kolmašová**[1,3], **Ondřej Lhotka**[1], **Jan Rusz**[1], **and Igor Strhárský**[2]

[1]Institute of Atmospheric Physics, Czech Academy of Sciences, Prague, 14100, Czech Republic
[2]Institute of Experimental Physics, Slovak Academy of Sciences, Košice, 04001, Slovakia
[3]Faculty of Mathematics and Physics, Charles University, Prague, 18000, Czech Republic

**Correspondence:** Jaroslav Chum (jachu@ufa.cas.cz)

**Abstract.** The cross-correlation between annual lightning frequency and solar activity and the heliospheric magnetic field (HMF) is examined on a global scale using corrected data from the World Wide Lightning Location Network (WWLLN) for the period 2009 to 2022. Relatively large regions with significant cross-correlation coefficients ($p < 0.05$) between the yearly lightning rates and sunspot number (SSN) are found in eastern Africa, part of South America overlapping with the South Atlantic Anomaly, and the Indian Ocean and west coast of Australia. The main region that shows a significant correlation between lightning activity and the $B_y$ component of the HMF and the magnetopause reconnection Kan–Lee electric field matches the South Atlantic Anomaly quite well. Also shown are areas that show a significant cross-correlation of lightning activity with the El Niño–Southern Oscillation index.

Similar areas of significant cross-correlation are obtained if simulated thunder days are used instead of lightning counts. Possible mechanisms leading to the observed correlations and limitations of the current study are discussed. The findings of the present study do not support previous works indicating that cosmic ray intensity is in phase with the global occurrence of lightning, but they do not rule out the role of cosmic rays in lightning ignition in developed thunderclouds and the role of energetic particles precipitating from the magnetosphere in the significant correlation between lightning and the $B_y$ component of the HMF (SSN) in the South Atlantic Anomaly.

## 1 Introduction

A possible relationship between solar activity and lightning–thunderstorm occurrence frequency has been investigated for many years. Fritz (1889) correlated thunderstorm frequencies with the sunspot number (SSN) for the period 1755–1875 and several European and North American stations without obtaining a conclusive result. A pioneering study on a global scale was made by Brooks (1934), who used data from 22 areas in different parts of the world and found that the cross-correlation coefficients between annual thunderstorm frequency and SSN were mostly positive. The best cross-correlation (0.88) was obtained for Siberia. However, this result was not confirmed by Kleymenova (1967). Brooks (1934) also showed that some cross-correlation coefficients varied considerably over relatively short distances or were relatively low (absolute value less than 0.2), for example in Europe. Other authors have studied the cross-correlation between thunderstorms and the solar cycle for specific regions. For example, Aniol (1952) investigated the solar influence on thunderstorm frequency in southern Germany over the interval 1881–1950 and found that the cross-correlation coefficients varied significantly for different subintervals. Stringfellow (1974) obtained a cross-correlation coefficient of 0.8 between thunderstorms in Great Britain and the solar cycle over the interval 1930–1973. Pinto Neto et al. (2013) identified the solar cycle in thunder day data obtained from selected Brazilian cities for the period 1951–2009 and found mostly an anti-phase relation between SSN and thunder day data.

The abovementioned past studies used daily records of audible thunder and did not deal with thunderstorm intensities

or the actual number of lightning strokes. This limitation can be overcome using lightning detection networks. Schlegel et al. (2001) calculated the cross-correlation coefficients between various parameters of solar activity and lightning detected in Germany and Austria using lightning detection systems for the period 1992–2000. In Germany, they found positive cross-correlation coefficients (around 0.8) between lightning and solar activity, but in Austria the results were inconclusive (cross-correlation coefficients close to zero). In addition, Schlegel et al. (2001) showed that cross-correlation coefficients might differ considerably when using lightning counts compared to using only the number of thunder days, as had been done in the past. A number of studies have also documented that lightning activity can be partially modulated on a shorter timescale by the solar rotation, the solar wind and the polarity of the heliospheric magnetic field (HMF; Chronis, 2009; Owens et al., 2014, 2015; Scott et al., 2014; Miyahara et al., 2018; Chum et al., 2021). Statistical studies by Voiculescu and Usoskin (2012) and Voiculescu et al. (2013) showed that solar activity might impact cloud cover in specific regions rather than globally.

The exact mechanism leading to the link between lightning and solar activity is unknown. Some authors believe that clouds, ionospheric potential and lightning activity might be modulated by the intensity of the cosmic ray (CR) flux entering the atmosphere. For example, Markson (1981) showed a positive correlation between the ionospheric potential (atmospheric electric field) and CR, which in turn is controlled by solar activity and HMF; the CR flux is anti-correlated with solar activity (Usoskin et al., 1998). Cosmic rays may influence lightning activity directly by providing secondary energetic particles (electrons) acting as a source of ionization necessary to ignite lightning, a process that is not yet understood in full detail (Dwyer and Uman, 2014; Shao et al., 2020). An indirect influence is based on the potential role of CR in the modulation of cloud electrification, cloud condensation nuclei and clouds (Markson, 1981; Kristjánsson et al., 2008; Kirkby, 2008; Svensmark et al., 2009). It is noted that a number of past studies (e.g., Brooks, 1934; Stringfellow, 1974; Schlegel et al., 2001) found mostly positive correlation between solar activity and lightning, implying a negative correlation with CR, which would reduce the importance of the direct ionization by CRs. Solar activity and weather/climate can also be linked through ultraviolet (UV) solar radiation, which is absorbed in the middle and upper atmosphere and strongly depends on solar activity. Changes in stratospheric temperature can then affect radiative balance, global circulation and potentially tropospheric weather (Gray et al., 2010). The exact mechanisms behind these changes need to be investigated. For example, the potential role of planetary waves in these top-down processes was discussed by Arnold and Robinson (1998, 2000) and Balachandran et al. (1999). Changes in the global electric circuit (GEC) associated with solar activity were discussed by Markson (1978), who put forward an idea that the atmospheric electricity is affected

by changes in column resistance above thunderstorms due to the ionizing radiation modulated by solar activity. This idea was further developed by Markson and Muir (1980) and Markson (1981) by investigating the relation between solar wind, cosmic rays and ionospheric potential and finding negative (positive) correlation between solar wind (cosmic rays). Hale (1979) suggested looking for effects more directly related to magnetospheric and auroral processes. On the other hand, Burns et al. (2008) and Lam and Tinsley (2016) have investigated the atmospheric electric field and associated pressure changes in polar regions and discussed the possible relationship between solar wind, namely the polarity of the $B_y$ component of the HMF, and tropospheric weather. They hypothesized that changes in the GEC, specifically through the downward current, could affect cloud microphysics, latent heat and cloud formation. However, further research and verification of this hypothesis are necessary. Voiculescu et al. (2013) showed that HMF partially affects cloud cover, specifically low cloud cover at middle and high latitudes, which could be consistent with HMF-driven changes in GEC, while it is possible that UV changes (a top-down mechanism) may play a more important role at low latitudes. Considerable attention has been paid to the chemical dynamical coupling caused by energetic particle precipitation (EPP) that includes both energetic electron precipitation from the radiation belt and solar proton events during enhanced geomagnetic and solar activity as a potential link between solar activity and climate. EPPs cause changes in the chemical composition of the mesosphere and stratosphere, leading to changes in radiative balance and atmospheric temperature (Seppälä et al., 2009; Andersson et al., 2014; Mironova et al., 2015; Sinnhuber et al., 2018). The role of planetary waves, the polar vortex and the phase of quasi-biennial oscillation in the effects of EPP on the atmosphere has often been discussed, with inconsistent results so far (Seppälä et al., 2013; Maliniemi et al., 2013, 2016; Salminen et al., 2019). Another hypothesis involving atmospheric waves was put forward by Prikryl et al. (2018), who, based on previous statistical studies, suggested that high-speed solar wind streams are, together with associated magneto-hydrodynamic waves, responsible for enhanced Joule heating in the high-latitude thermosphere and ionosphere, which in turn generates atmospheric gravity waves that propagate equatorward and may reach the troposphere, lift the air, and initiate convection and cloud formation.

The above review of possible coupling mechanisms indicates that further experimental and theoretical studies are needed to evaluate the relative role and validity of different mechanisms that may link solar activity to climate and lightning frequency. The present study investigates the relation between solar activity (SSN), the $B_y$ and $B_z$ components of the HMF, CR, and lightning activity in various regions around the globe using the World Wide Lightning Location Network.

## 2  Measurement setup and methods

The near-Earth solar wind data and data of solar activity were retrieved from the NASA Goddard Space Flight Center (GSFC) Space Physics Data Facility OMNIWeb Service (https://omniweb.gsfc.nasa.gov/form/, last access: 3 February 2023). The solar data were also compared with the CR flux measured by a neutron monitor (NM) with the cutoff rigidity of 3.84 GV located on the summit of Lomnický štít (49.195° N, 20.213° E) at an altitude of 2634 m. The NM is filled with $BF_3$ and is of type NM64. More information about the NM can be found in Kudela and Langer (2009) and Chum et al. (2020).

Global lightning data were obtained using the World Wide Lightning Location Network (WWLLN), which consists of approximately 70 sensors operating in the frequency range of 3–30 kHz and receiving electromagnetic signals that are generated by lightning strokes and propagate in the waveguide between the Earth's surface and the lower ionosphere (Rodger et al., 2004). The WWLLN was selected because of its global coverage and availability for the authors. It should be noted that the optical Lightning Imaging Sensor (LIS) detector on the satellite observes mainly low latitudes and that the Optical Transient Detector (OTD) with global coverage worked only from 1995 to 2000. The WWLLN lightning counts in $1° \times 1°$ bins are used in this study, but it is also shown that similar results are obtained if larger bins (3° latitude $\times$ 6° longitude) are used. The data available to the authors started in 2009. It should also be noted that the number of WWLLN sensors was substantially lower before 2009, and therefore the detection efficiency was also significantly lower than today. In addition, corrections of detection efficiency (used in this study and described later) are not available for data from before 2009. It is estimated that the current detection efficiency for cloud-to-ground (CG) strokes with a peak current of at least 30 kA is approximately 30 % globally (http://wwlln.net/, last access: 31 January 2024).

To investigate the possible dependence of lightning activity on the solar cycle, we applied a cross-correlation analysis using 1-year lightning counts and 1-year averages of sunspot number, NM counts, and $B_y$ and $B_z$ components of HMF in the Geocentric Solar Ecliptic (GSE) coordinate system. The 1-year values were used to remove the seasonal dependence of lightning occurrence. The lightning frequency trends are shown over the 2009–2022 interval. The trend in lightning data is likely caused by increasing network efficiency due to the increasing number of WWLLN sensors. The dependence of the number of detected lightning strokes on the number of WWLLN sensors was shown by Holzworth et al. (2021). Their Fig. 2 shows a clear decrease in the number of lightning detections before $\sim$ 2013 due to the lower number of sensors. The applicability of the WWLLN was also discussed by Virts et al. (2013), who verified that WWLLN lightning climatology is consistent with in situ rain observations. Hutchins et al. (2012) introduced a model that account for the uneven global coverage of the WWLLN sensors and variations in the propagation of very-low-frequency (VLF) signals using correction coefficients for detection efficiency, currently provided for each hour and $1° \times 1°$ bin. The correction is especially large for Africa due to the low number of sensors. As will be shown in the Results section, this model (correction) gives relatively high lightning frequency in Africa during the period $\sim$ 2009–2013. Therefore, results are also presented for the uncorrected data.

To compare time series with different units, scales and relative fluctuations, it is useful to standardize data (normalized by standard deviation after subtracting the mean) using Eq. (1).

$$a_{\text{norm}} = \frac{a - \text{mean}(a)}{\sigma_a}, \tag{1}$$

where $a$ is the analyzed quantity (lightning counts, SSN, components of HMF, NM counts, etc.) and $\sigma_a$ is the standard deviation of its distribution. The cross-correlation coefficients $c$ are calculated as follows:

$$c = \frac{1}{N-1} \cdot \sum_{i=1}^{N} \frac{a - \text{mean}(a)}{\sigma_a} \frac{b - \text{mean}(b)}{\sigma_b}$$

$$= \frac{1}{N-1} \cdot \sum_{i=1}^{N} a_{\text{norm}} \cdot b_{\text{norm}}. \tag{2}$$

The statistical significance is obtained using $t$ statistics for $N - 2$ degrees of freedom and calculated by the corrcoef function in MATLAB software.

To compare the cross-correlation coefficients obtained for lightning frequency with those for thunder days (a parameter used in many previous studies), we estimate the thunder days for each bin. The thunder days are estimated as follows. First we calculate the ratio ($r_{\text{LAT}}$) of the area of the $1° \times 1°$ bin ($A_{\text{LAT}}$) to the thunder detection area ($A_T$), considering the dependence of the bin area on latitude. The thunder detection area is computed as $\pi \rho^2$, where $\rho = 20$ km, which is the middle value of the thunder audibility range (15–25 km) given by Pinto et al. (2013). The value of ratio $r_{\text{LAT}}$ is largest at the Equator (9.86) and decreases with increasing latitude (e.g., it is 5 at the latitude of 50°). Then, to allow some uncertainty in the thunder days (TD), the TD are not determined from a fixed threshold $r_{\text{LAT}}$ but are simulated using logistic function and summed over the year to obtain annual values:

$$\text{TD} = \sum_{i=1}^{M} \frac{1}{1 + e^{-(N_i - r_{\text{LAT}})}}, \tag{3}$$

where $N_i$ is the number of lightning detections in the specific bin on the $i$th day and $M$ is the number of days in a year. The logistic function (the individual term summed in Eq. 3) is very close to zero for $N_i \ll r_{\text{LAT}}$ and approaches 1 if $N_i \gg r_{\text{LAT}}$. A relatively narrow range of intermediate values of the logistic function around $N_i \approx r_{\text{LAT}}$ admits some uncertainties.

The potential influence of El Niño–Southern Oscillation (ENSO) on thunderstorm occurrence (Williams et al., 2021; Kolmašová et al., 2022) is also investigated by calculating cross-correlation coefficients between yearly lightning counts (thunder days) and the yearly mean of the ENSO index. The ENSO index was taken from the following NASA web page: https://sealevel.jpl.nasa.gov/data/vital-signs/el-nino/ (last access: 2 May 2024).

It should be noted that the solar wind electric field components $E_{zSW} \sim -v_x B_y$ and $E_{ySW} \sim v_x B_z$ are believed to penetrate and add to the Earth's internal electric field between the ionosphere and ground (Rycroft et al., 2000; Lam and Tinsley, 2016), but since the relative changes in $B_y$ and $B_z$ are much larger than the relative changes in the Earthward solar wind speed $v_x$, only the dependencies of the Earth's electric field on $B_y$ or $B_z$ have been frequently studied (e.g., Burns et al., 2008). We verified that differences between the results obtained for $|v_x|B_y$ ($|v_x|B_z$) and $B_y$ ($B_z$) are negligible.

In addition, the cross-correlation is also computed between lightning counts and the magnetopause reconnection electric field (Kan and Lee, 1979). This electric field, namely its perpendicular component, can serve as a proxy for ionospheric electric currents at high latitudes (namely Region 1) during geomagnetic storms; potential across a polar cap (Kan and Lee, 1979; Mannucci et al., 2014); or large-scale traveling ionospheric disturbances (LSTIDs) – waves in the upper atmosphere and ionosphere (Borries et al., 2023). The perpendicular component (related to the magnetic field lines at the magnetopause) of the Kan–Lee electric field (Kan and Lee, 1979) is

$$E_{perp} = v_x B_T \sin^2(\varphi/2), \tag{4}$$

where $B_T$ is $\sqrt{(B_y^2 + B_z^2)}$ and $\varphi$ is a clock angle of the transverse HMF (relative to the $z$ axis): $\varphi = \mathrm{atan}(B_y/B_z)$.

The parallel component of the electric field is often neglected in plasma physics because it is believed that it is usually small because of high conductivity along the field line, but Kan and Lee (1979) also pointed out that the parallel component of the reconnection electric field ($E_{par}$) exists and should not be automatically neglected. The parallel field might accelerate/decelerate particles along the field line and affect their trajectories and precipitation into the atmosphere.

$$E_{par} = v_x B_T \sin(\varphi/2)\cos(\varphi/2) \tag{5}$$

## 3   Results

Figure 1a shows a world map with the global distribution of the total corrected number of lightning strokes recorded by the WWLLN during the analyzed period of 2009–2022. The color scale indicates the common logarithm of lightning strokes in each $1° \times 1°$ bin for the latitude range from $-66$ to $66°$. Figure 1b shows, for comparison, the global distribution of the simulated thunder days using Eq. (3). Thunderstorm

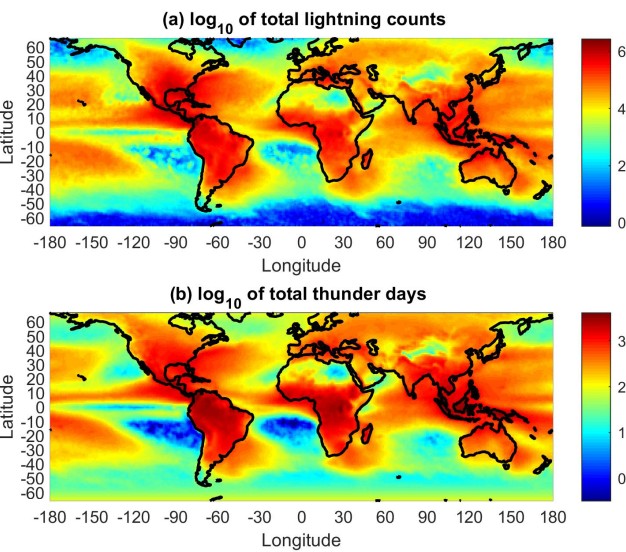

**Figure 1.** Common logarithm of corrected numbers of all lightning strokes **(a)** and all thunder days **(b)** during the analyzed period, 2009–2022.

centers are readily verified in tropical and subtropical regions over the continents, namely central Africa, South and Central America, East Asia, and Indonesia. The continental lightning dominates the oceanic lightning by more than an order of magnitude. Significant numbers of lightning strokes are also recorded in the Mediterranean. It should be noted that the actual number of lightning strokes is larger because of the limited detection efficiency of the WWLLN, especially for intracloud discharges. Compared to the LIS–OTD climatology data set (https://ghrc.nsstc.nasa.gov/lightning/data/data_lis_otd-climatology.html, last access: 13 February 2024), the WWLLN underestimates the lightning frequency, especially in central Africa, where the number of uncorrected lightning strokes detected by the WWLLN is about 10 times lower. Therefore, the applied corrections (mentioned in the previous section) are the largest in Africa, as will also be shown later.

The cross-correlation coefficients between the yearly SSN and corrected yearly lightning counts are shown in Fig. 2a. The cross-correlation coefficients are displayed only for those bins for which the correlation (anti-correlation) is statistically significant (probability of null hypothesis; $p < 0.05$) and the total number of detected lightning strokes was larger than $2 \times 10^3$ for the entire period 2009–2022, which corresponds to an average yearly number of detected lightning strokes larger than $\sim 140$ in each bin. The same threshold for the required number of detected lightning strokes per bin is used in the following analogous figures. A red color indicates cross-correlation coefficients close to 1, whereas dark blue stands for large negative values of cross-correlation coefficients. It is obvious that lightning activity is in phase – it correlates well with solar activity represented by SSN in cen-

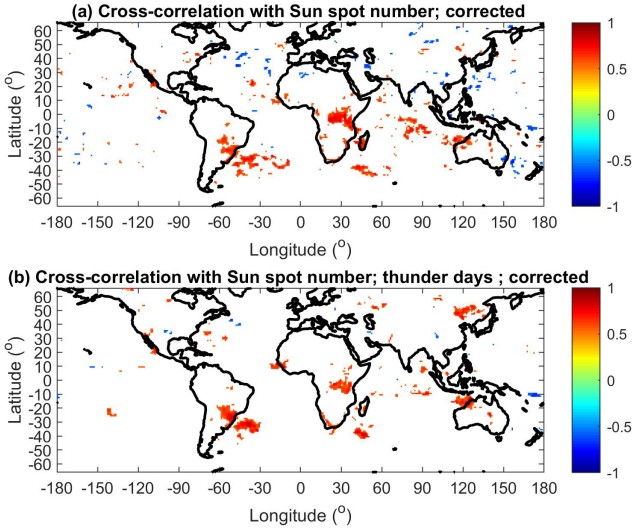

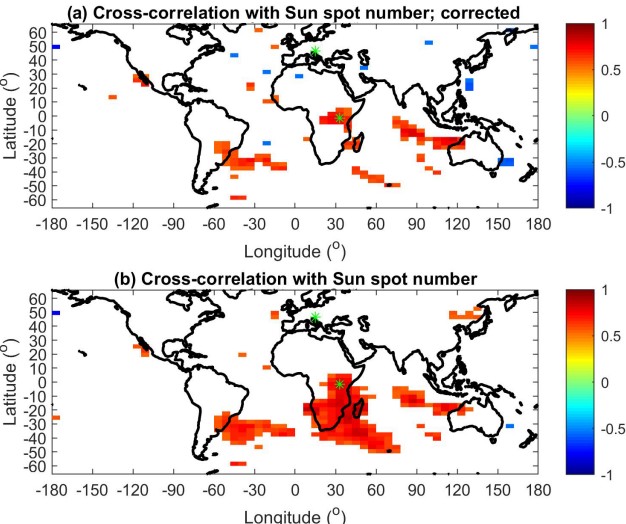

**Figure 2. (a)** Cross-correlation coefficients between the yearly SSN and corrected number of lightning strokes in $1° \times 1°$ bins. **(b)** Cross-correlation coefficients between the yearly SSN and simulated yearly thunder days in $1° \times 1°$ bins. Only statistically significant cross-correlation coefficients are displayed ($p < 0.05$).

**Figure 3. (a)** Cross-correlation coefficients between the yearly SSN and corrected number of lightning strokes in $3° \times 6°$ (latitude × longitude) bins. **(b)** Cross-correlation coefficients between the yearly SSN and uncorrected number of lightning strokes in $3° \times 6°$ (latitude x longitude) bins. Only statistically significant cross-correlation coefficients are displayed ($p < 0.05$). The green asterisks indicate the locations of the selected bins for which time series are shown in Fig. 4.

tral and eastern Africa, part of South America, and the South Atlantic Anomaly region and west coast of Australia.

For comparison with previous works, it is also useful to investigate which regions would exhibit significant cross-correlation coefficients if thunder day data were used. We use simulated thunder day data obtained from the corrected WWLLN lightning counts by the method described in Sect. 2. The required threshold of $2 \times 10^3$ lightning strokes for each bin was modified to $2 \times 10^2$ thunder days, and the $1° \times 1°$ bins were used. Figure 2b displays the cross-correlation coefficients between the yearly sunspot number and simulated yearly thunder days. Although the exact shape of the main regions that show a significant correlation is partly different from that of the regions in Fig. 2a, which shows the same as Fig. 2b but using the number of lightning strokes, their approximate locations remain the same: eastern Africa, part of South America and the west coast of Australia. In addition, there is a relatively large region in East Asia that exhibits significant correlation if thunder days are used.

Figure 3a shows the cross-correlation coefficients between the yearly SSN and corrected yearly lightning counts in $3°$ latitude $\times 6°$ longitude bins to demonstrate that the main centers of significant correlation do not change if a different bin size is used (compare Figs. 2a and 3a). Figure 3b displays the cross-correlation coefficients between the yearly SSN and uncorrected yearly lightning counts in $3°$ latitude $\times 6°$ longitude bins to show the effect of correction on WWLLN data. The largest difference is in Africa, where the area with significant correlation is much larger for the uncorrected data. The reason for that is clear from the time series that are presented in Figs. 4 and 6.

Figure 4 displays the time series of the annual SSN (Fig. 4a) and annual averages of NM counts measured at Lomnický Štít each minute (Fig. 4b). The relative deviations of the NM data from their means are much smaller than for SSN. The mean value and standard deviation for SSN are 47.6 and 38.4, respectively, and for NM counts they are 27 691 and 842. Obviously, the time series of the SSN and NM data are in anti-phase (anti-correlated). This is expected since it is known that the CR flux characterized by NM data is anti-correlated with solar activity (e.g., Usoskin, 1998). An example of the time series of the annual number of lightning strokes for the selected bin in eastern Africa (latitude from $-3$ to $0°$ and longitude from 30 to $36°$), in which relatively high and significant cross-correlation coefficients (0.77 for corrected data and 0.90 for corrected smoothed data TS1 CE1) were obtained, is shown in Fig. 4c. Blue represents the uncorrected numbers of lightning strokes, and red represents the corrected numbers using the provided correction coefficients of detection efficiency. The uncorrected and corrected lightning counts significantly differ before 2014. The corrected data are relatively high before 2014, when the solar activity is lower. This is even more remarkable in the surrounding bins and leads to a smaller region of significant correlation compared to the results obtained for uncorrected data (compare Fig. 3a and b). On the other hand, in most of the other regions, such as in the selected bin shown in Fig. 3d (latitude from 45 to $48°$ and longitude from 12 to $18°$), in which the cross-correlation is statistically insignificant, the differ-

**Table 1.** Cross-correlation coefficients $C_{\text{SSN},i}$ between the yearly SSN and NM data, the $B_y$ and $B_z$ components of HMF, the reconnection electric field, and the ENSO index with the corresponding $p$ values.

|  | NM | $B_y$ | $B_z$ | $E_{\text{perp}}$ | $E_{\text{par}}$ | ENSO |
|---|---|---|---|---|---|---|
| $C_{\text{SSN},i}$ | −0.94 | 0.34 | 0.17 | 0.68 | 0.28 | −0.42 |
| $p$ value | $< 10^{-6}$ | 0.24 | 0.55 | 0.007 | 0.33 | 0.14 |

ences between corrected and uncorrected lightning counts are small, as shown in Fig. 4d. The selected bins are marked by green asterisks in Fig. 3. Figure 5 shows the time series of annual values of the $B_y$ and $B_z$ components of HMF, the perpendicular and parallel components of the Kan–Lee electric field calculated from 1 d values, and yearly means of the ENSO index. The normalized annual time series of SSN, the NM and lightning counts for the selected bins are presented in Fig. 6.

As discussed in the Introduction, some previous studies showed a relation between the polarity (sign) of the HMF components (especially of $B_y$) and the atmospheric electric field at high latitudes and lightning or cloud cover at specific altitudes (Burns et al., 2008; Voiculescu et al., 2013; Owens et al., 2014). First, it is useful to investigate how the individual components of the HMF correlate with SSN. The cross-correlation coefficients between the yearly NM data used, HMF components, Kan–Lee reconnection electric field and ENSO index are shown in Table 1.

The NM data are very well anti-correlated ($-0.94$, $p < 10^{-6}$) with SSN, so maps of cross-correlation coefficients between NM and lightning counts just give an opposite (negative) image to the maps shown, e.g., in Fig. 2. More interesting is a map of cross-correlation coefficients between the $B_y$ and $B_z$ components of the HMF and lightning counts, shown in Fig. 7. It is obvious that lightning activity correlates with $B_y$ over the southeastern part of South America (including the South Atlantic) and over smaller regions in Europe, Asia and North America (Fig. 7a). On the other hand, only a few relatively small regions show significant cross-correlation between lightning counts and $B_z$ (Fig. 7b). A comparison of Figs. 2a and 7a reveals that the main difference between maps for the cross-correlation with SSN and the $B_y$ component is that significant cross-correlation with $B_y$ is not found in Africa. Figure 8 shows that similar results are obtained if thunder days, instead of lightning counts, are used. In addition, regions that show anti-correlation (e.g., in Colombia and Venezuela) are identified in Fig. 8a. Again, practically no significant cross-correlation is found with the $B_z$ component (Fig. 8b).

The cross-correlation coefficients between the corrected lightning counts and magnetopause reconnection electric field (Kan–Lee) are shown in Fig. 9. Significant and large values (up to about 0.85) are obtained in the southeastern

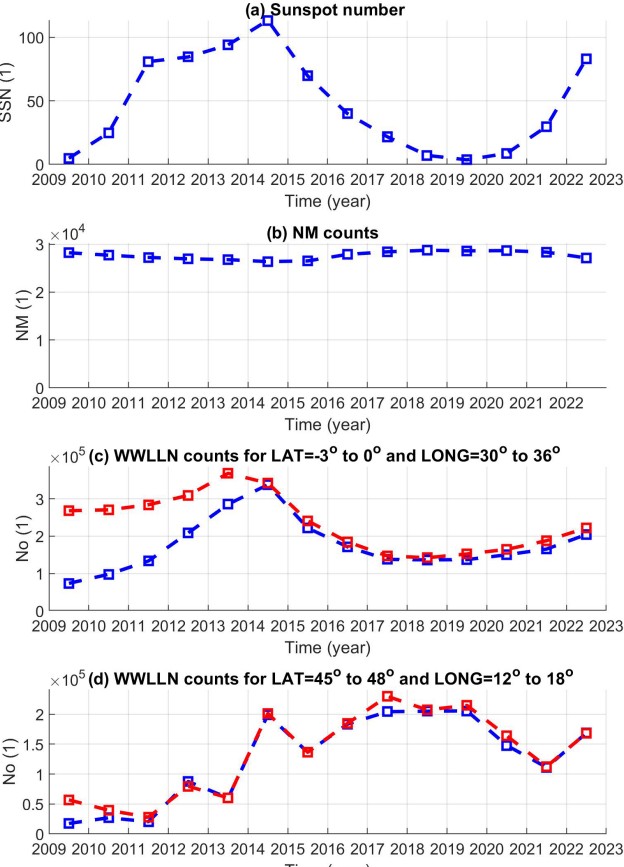

**Figure 4. (a)** Yearly sunspot number. **(b)** Yearly averages of 1 min NM counts measured at Lomnický Štít. **(c)** Number of detected lightning strokes in the selected bin in which high cross-correlation with SSN was found, latitude from $-3$ to $-0°$ and longitude from 30 to 36°. **(d)** Number of detected lightning strokes in the selected bin in which significant correlation with SSN was not found, latitude from 45 to 48° and longitude from 12 to 18°. The corrected lightning counts are in red (see text for more details).

part of South America for both the perpendicular component (Fig. 9a) and the parallel component (Fig. 9b). The map of cross-correlation coefficients, mainly for the parallel component of the Kan–Lee electric field, is very similar to the map for cross-correlation with $B_y$ (compare Figs. 7a and 9b). This is actually not surprising because $|B_y|$ is usually larger than $|B_z|$. Therefore $B_T \approx |B_y|$ in the first approximation, and since, in addition, the absolute value of the term $\sin(\varphi/2)\cos(\varphi/2)$ peaks for $B_z = 0$ ($|\varphi|/2 = 45°$), the fluctuations in the parallel component of the Kan–Lee electric field defined by Eq. (5) roughly follow the fluctuations in $B_y$, including the sign. Similar results are obtained if thunder days are used.

Williams et al. (2021) and Kolmašová et al. (2022) showed that ENSO can influence lightning occurrence. Figure 10 shows maps of significant ($p < 0.05$) cross-correlation coefficients between the ENSO index and corrected lightning

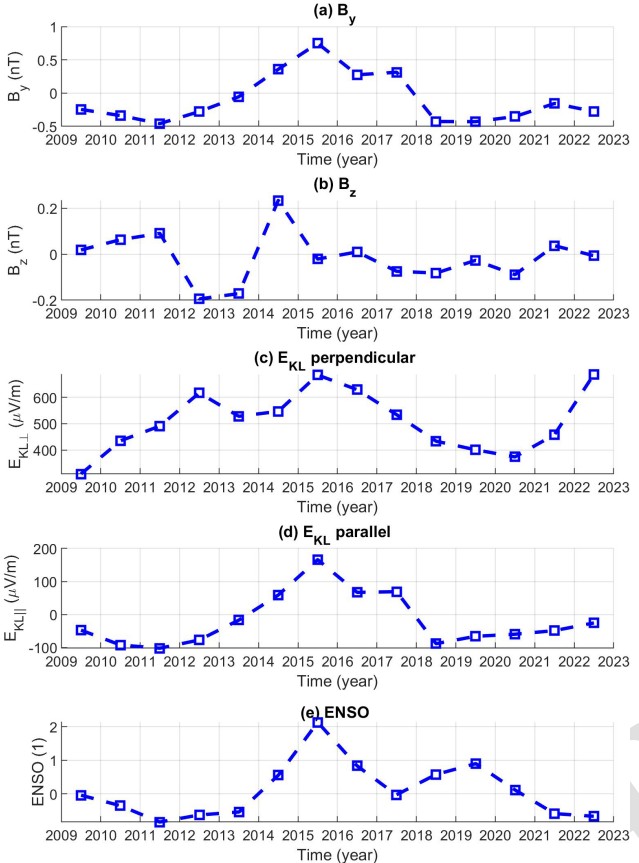

**Figure 5.** Yearly means of $B_y$ **(a)**, $B_z$ **(b)**, the perpendicular **(c)** and parallel **(d)** components of the Kan–Lee electric field calculated from 1 d values, and the ENSO index **(e)**.

counts (Fig. 10a) and simulated thunder days (Fig. 10b). The results for lightning counts and thunder days are qualitatively similar, but regions of negative (anti-phase) correlation are larger if thunder days are used, especially in equatorial America, equatorial Atlantic and Indonesia. Negative cross-correlations are also identified in southern Africa. Positive correlations are found not only in South America and the eastern Pacific, but also partly in Europe, the Mediterranean and the western part of the USA.

## 4 Discussion and conclusions

The presented maps show that significant cross-correlation coefficients ($p < 0.05$) between solar activity represented by SSN and lightning are observed in central and eastern Africa; the southeastern part of South America, including part of the South Atlantic; and the west coast of Australia and part of the Indian Ocean for the period 2009–2022. It should be noted that the regions of significant correlation do not include most of the typical wet rainforest areas: the Amazon basin in South America, the west part of the Congo basin in Africa, and Southeast Asia and Indonesia. The total area showing signif-

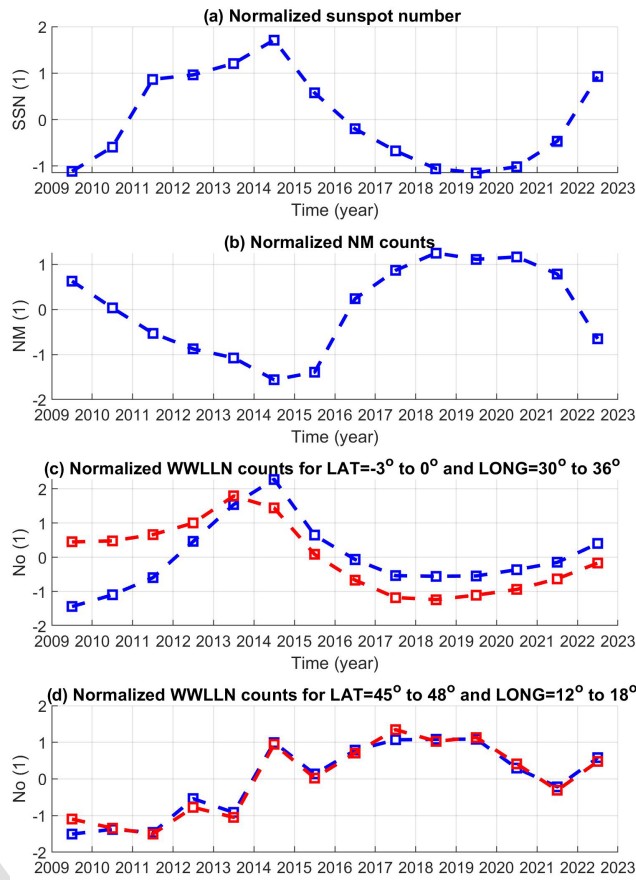

**Figure 6. (a)** Normalized yearly SSN. **(b)** Normalized yearly NM counts measured at Lomnický Štít. **(c)** Normalized number of lightning strokes in the selected bin in which significant cross-correlation with SSN was found, latitude from $-3$ to $-0°$ and longitude from 30 to 36°. **(d)** Normalized annual number of lightning strokes in the selected bin in which significant correlation with SSN was not found, latitude from 45 to 48° and longitude from 12 to 18°. Corrected normalized counts are in red, and uncorrected counts are in blue.

icant cross-correlation is relatively small, and a random coincidence, at least in part, cannot be entirely ruled out. Therefore, we discuss possible mechanisms or connections below. In this respect, it should also be noted that the areas with significant cross-correlation between lightning activity and the $B_z$ component of the HMF are especially small and randomly distributed. In this latter case, a random coincidence is quite probable.

Mutai and Ward (2000) found that rain events in eastern Africa are associated with the Madden–Julian Oscillation (MJO) in the Indian Ocean. Rain in Africa is usually associated with thunderstorms. Kozlov et al. (2023) found that the ionospheric potential follows the MJO phase. We note that both eastern Africa and the Indian Ocean show significant correlation with SSN. It was also shown that the intensities of rain events in eastern Africa depend on the phase

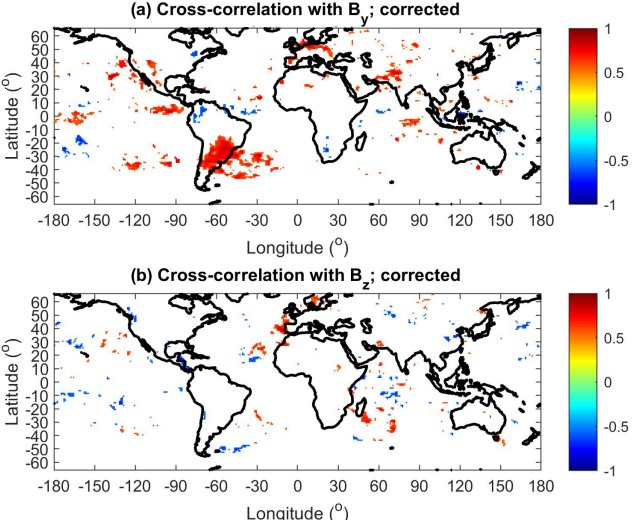

**Figure 7. (a)** Cross-correlation coefficients between the yearly $B_y$ component of HMF and corrected lightning counts in $1° \times 1°$ bins. **(b)** Cross-correlation coefficients between the yearly $B_z$ component of HMF and corrected lightning counts in $1° \times 1°$ bins. Only statistically significant cross-correlation coefficients are displayed ($p < 0.05$).

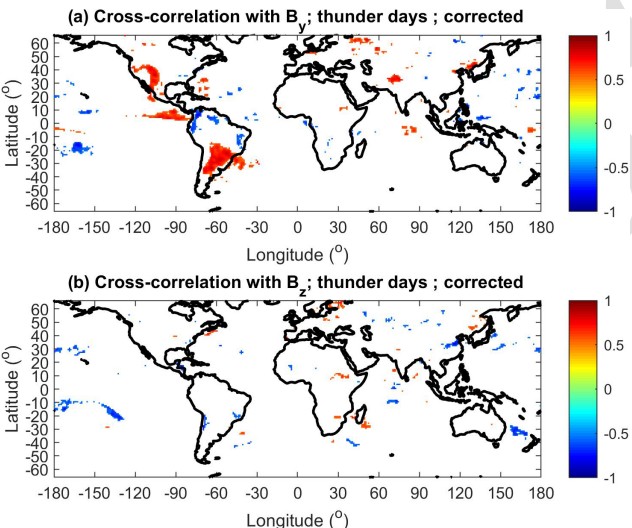

**Figure 8. (a)** Cross-correlation coefficients between the yearly $B_y$ component of HMF and corrected lightning counts in $1° \times 1°$ bins. **(b)** Cross-correlation coefficients between the yearly $B_z$ component of HMF and corrected lightning counts in $1° \times 1°$ bins. Only statistically significant cross-correlation coefficients are displayed ($p < 0.05$).

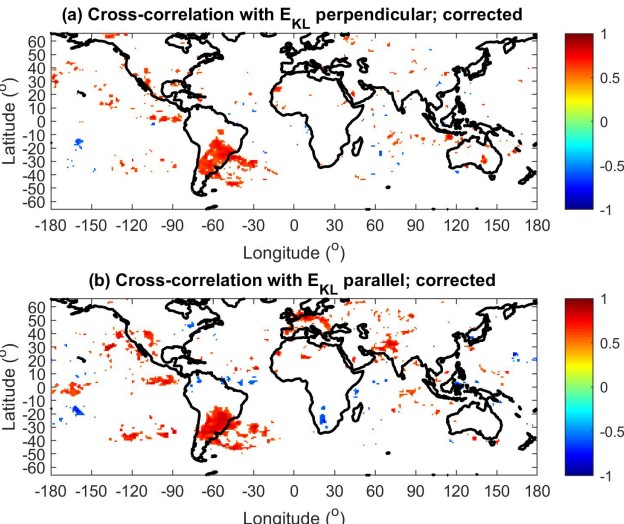

**Figure 9. (a)** Cross-correlation coefficients between the reconnection Kan–Lee electric field (perpendicular component) and corrected lightning counts in $1° \times 1°$ bins. **(b)** Cross-correlation coefficients between the reconnection Kan–Lee electric field (parallel component) and corrected lightning counts in $1° \times 1°$ bins. Only statistically significant cross-correlation coefficients are displayed ($p < 0.05$).

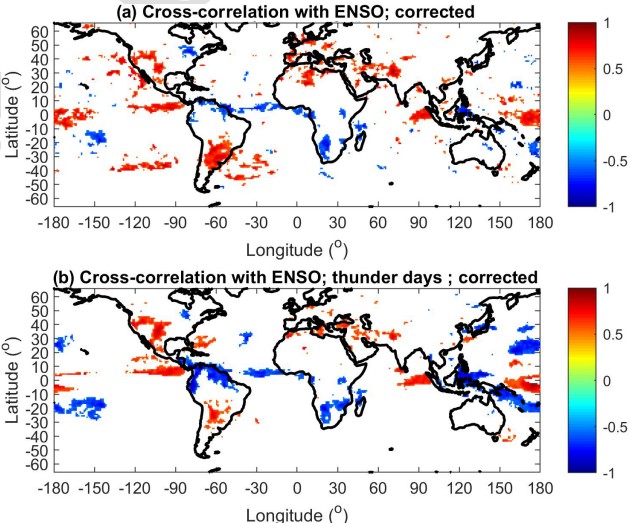

**Figure 10. (a)** Cross-correlation coefficients between the yearly mean of the ENSO index and the corrected number of lightning strokes in $1° \times 1°$ bins. **(b)** Cross-correlation coefficients between the yearly mean of the ENSO index and simulated thunder days in $1° \times 1°$ bins. Only statistically significant cross-correlation coefficients are displayed ($p < 0.05$).

of ENSO (Ogallo, 1988; Nicholson and Kim, 1997). However, our analysis does not show significant cross-correlation between the lightning activity and ENSO index in eastern Africa (Fig. 10). Pinto et al. (2013) found that an increase in thunderstorm activity around Rio de Janeiro occurs simultaneously with a positive anomaly of the South Atlantic

sea surface temperature and La Niña. This is partly confirmed by our study; however, the regions that show significant anti-phase correlation with the ENSO index are rather the equatorial regions of South America (Fig. 10). Williams et al. (2021), based on Schumann resonance measurements,

found that global lightning activity increased in the transition phase to El Niño. It should be noted that a strong El Niño phase occurred in 2014–2015, which coincided with a maximum of solar cycle 24. On the other hand, there were two relatively weak El Niño phases during the 2009–2010 and 2018–2019 solar minima, but the 2009–2010 El Niño in particular was very short and difficult to recognize in the annual averages of the ENSO index (Fig. 5e). Nevertheless, it should be noted that the cross-correlation maps of lightning activity with SSN (Fig. 2) and the ENSO index (Fig. 10) look different; in the latter case, more regions with significant anti-phase relation can be found.

It is probable that regional climatic differences are responsible for the observed patterns in the maps of significant correlation (anti-correlation) with SSN, but the exact mechanism is not clear and needs further investigation. Barriopedro et al. (2008) found that the 11-year solar cycle (represented by SSN) modulates atmospheric blocking in mid-latitudes of the Northern Hemisphere. Proposed underlying physical mechanisms are related to heating in the stratosphere by UV radiation (Gray et al., 2016). Changes in blocking frequency, persistence and locations affect atmospheric circulation, which is a main factor modulating surface weather and climate patterns at mid-latitudes (Masato et al., 2012). However, the observed relationships between SSN and atmospheric circulation in the troposphere were significant only in boreal winter, when the lightning activity was relatively low compared to in summer, and cannot explain the relation based on annual data.

Pinto Neto et al. (2013), using audible thunder data from 1951 to 2009, found a solar cycle signature in thunderstorm activity in several Brazilian cities, with a significant anti-phase relation with SSN for three out of seven cities. This is not confirmed in the current study, which covers the period from 2009 to 2022. In contrast, a significant in-phase relation was found in the southern part of Brazil. It should be noted that the anti-correlation is consistent with the idea of Markson (1981), who suggested that thunderstorm activity is in phase with cosmic rays and in anti-phase with solar activity. According to our study, however, cosmic rays are uncorrelated with lightning activity over most of the globe. Probably more important are weather conditions leading to thunderstorm formation. This does not rule out the possibility of cosmic rays playing a role in igniting individual lightning strikes in thunderclouds that have already developed (Shao et al., 2020).

Comparison of the maps obtained using lightning counts and simulated thunder days shows that although the patterns of areas with significant cross-correlation with SSN are not exactly identical, they are not very different and the approximate location of the major centers remains the same. Unlike Schlegel et al. (2001), we have not found significant correlation between the lightning frequency in Germany and SSN. On the other hand, a significant correlation was found between lightning frequency in Germany and the $B_y$ component of the HMF and the reconnection Kan–Lee electric field (Figs. 7a, 9b).

An important and interesting result of the present study is that the region of significant correlation between lightning activity and the $B_y$ component of the HMF and the reconnection Kan–Lee electric field coincides with the region of the South Atlantic Anomaly (SAA). This result is valid for both lightning counts and thunder days. It is known that a relatively large number of energetic particles precipitate from the magnetosphere into the atmosphere due to the decreased strength of the magnetic field in the SAA region, especially during the interaction of solar wind with the Earth's magnetosphere. For example, Sauvaud et al. (2008) showed, using measurements taken on board the DEMETER satellite, a large flux of 200 keV precipitating (loss-cone) electrons when the satellite was inside the SAA. On the other hand, eastern Africa does not exhibit correlation with the $B_y$ component of the HMF and the reconnection Kan–Lee electric field. Therefore, the possibility of different mechanisms being responsible for the significant correlation between SSN and lightning activity in the SAA region and in eastern Africa, where the precipitation of energetic particles from the magnetosphere is unlikely, cannot be excluded. Further studies are needed to verify whether energetic particles precipitating from the magnetosphere are indeed responsible for the significant correlation between lightning activity and the Kan–Lee electric field ($B_y$ component of the HMF) in the SAA region. The energy spectrum of precipitating particles, their effect on ionization, electric conductivity, chemical compositions at different heights, radiative balance, cloud cover and cloud charging need to be analyzed.

It should also be noted that precipitating particles and solar X-rays, which are usually stronger during solar maxima, enhance ionization in the bottom ionosphere and upper mesosphere, lowering the reflection height from which the very-low-frequency and extra-low-frequency electromagnetic waves are reflected (Sátori et al., 2005; Bozóki et al., 2021). Changes in properties in the upper part of the Earth–ionosphere waveguide can thus bias/affect the detection efficiency of the WWLLN.

Another limitation of the current study is a relatively short analyzed period, 2009–2022, which only covers solar cycle 24 and the beginning of solar cycle 25. In addition, the time series variations in the $B_y$ component of the HMF and ENSO index look very similar before 2017 (Fig. 5a and e). It is not clear whether the patterns obtained of significant cross-correlation coefficients between lightning and solar activity or HMF are also valid for other time periods/solar cycles. Some previous studies based on thunder days, such as that of Aniol (1952), suggest that the cross-correlation coefficients between thunder days in Germany and solar activity vary with time. Similarly, Chum et al. (2021) identified a period of solar rotation in lightning data in central Europe in 2016–2019 (lightning was more probable if the HMF was oriented toward the Sun). However, extending their study to a

longer time interval reveals that the period observed in lightning data and the period of solar rotation (period of HMF polarity) are generally asynchronous, although they may be close together. Further studies, based on longer time intervals, are needed to verify the results presented in this study.

## Appendix A: List of abbreviations

| Abbreviation | Description |
| --- | --- |
| CRs | Cosmic rays |
| ENSO | El Niño–Southern Oscillation |
| HMF | Heliospheric magnetic field |
| MJO | Madden–Julian Oscillation |
| NM | Neutron monitor |
| WWLLN | World Wide Lightning Location Network |
| SAA | South Atlantic Anomaly |
| SSN | Sunspot number |

**Data availability.** WWLLN archival data are copyrighted by the University of Washington and are available to the public at nominal cost. The solar activity and HMF data can be found at the NASA GSFC Space Physics Data Facility OMNIWeb Service (https://omniweb.gsfc.nasa.gov/, NASA, 2024). The NM data can be downloaded from http://data.space.saske.sk/status/ (Balaz and Strharský, 2017; access can be provided by Ronald Langer (langer@saske.sk) on request).

**Author contributions.** JC designed and wrote the paper and performed most of the analysis. RL and IS are responsible for and provided the SCR data. IK provided the lightning data and contributed to the discussion. OL and JR contributed to the discussion. All authors read and approved the submitted version.

**Competing interests.** The contact author has declared that none of the authors has any competing interests.

**Disclaimer.** Publisher's note: Copernicus Publications remains neutral with regard to jurisdictional claims made in the text, published maps, institutional affiliations, or any other geographical representation in this paper. While Copernicus Publications makes every effort to include appropriate place names, the final responsibility lies with the authors.

**Acknowledgements.** We are grateful to Samuel Štefánik for maintaining the measurements on Lomnický Štít. The authors thank Earle Williams and the anonymous reviewer for valuable comments that greatly improved the manuscript.

**Financial support.** Support was provided by the Czech Academy of Sciences (grant no. SAV-23-02). The work of Ivana Kolmašová was supported by the Czech Science Foundation (grant no. 23-06430S).

**Review statement.** This paper was edited by Graciela Raga and reviewed by Earle Williams and one anonymous referee.

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

## Remarks from the language copy-editor

## Remarks from the typesetter