# Peer review of "Solar cycle signatures in lightning activity"

_EGUsphere, 2023_

## Author Response (AR1)

**Review of Solar Cycle Signatures in Lightning Activity, by J. Chum, R. Langer, I. Kolmasova, O. Lhotka, J. Rusz and I. Strharsky**

This study revisits an important area of research with a very mixed collection of findings in earlier work. The initiative to work with lightning strokes rather than thunder days (the traditional approach) may represent an improvement, but the selection of the WWLLN data set to do the stroke test over a single solar cycle has notable limitations. Apparent contradictions with earlier findings need to be addressed, and broader attention to the literature on this topic, and especially in the realm of the global circuit of atmospheric electricity, is needed. Substantial improvements will be needed to move this work to publication.

Thank you for your careful reading of the manuscript and for your thoughtful and critical comments. We believe they have contributed to the improvement of the manuscript.

General response. We carefully considered the comments and significantly changed the analysis and rewrote substantial part of the manuscript. The main changes are:

- The analysis is now based on the corrected WWLLN counts using the official correction coefficients to account for changes in detection efficiency (instead of previously used questionable detrending using quadratic function).
- A comparison with thunder days (simulated) used in previous studies is also provided.
- Only areas with statistically (95%) significant correlation (anti-correlation) are now shown in the maps, i.e. for p<0.05.
- The changes led to partially (not completely) different results. Accordingly, the text and discussion was changed. For example, the significant correlation in the region of South Atlantic anomaly is more clear. Also, it can be seen that most of the tropical rain forests are not significantly correlated.

**Summary: Consider for publication after major revision**

1) Comments on main findings in Figure 2a

At the outset, the results in Figure 2a, the most important figure in the manuscript, are rather amazing in showing such large and coherent positive correlation coefficients over two of three major lightning zones. My first reaction; if this is a meaningful indication of a positive relationship between the sunspot representation of solar activity and lightning activity on the 11-year time scale, why hasn't this been exposed in earlier studies? By 'earlier' studies I can be more specific in sections 2 and 7 below.

I have some questions about general procedure. Why were quadratic fits implemented? How do they affect the results? Normalizing the data can affect the amplitude and the cross correlation. This should be considered. (You did not show the raw data for Figure 3.) No information is included here about statistical significance. The availability of a single solar cycle in the analysis should have an impact on significance. What about p values? What about lag results in the important context of phase, and see further below. No frequency analysis is included.

The generality "correlation does not guarantee causality" is also deserving of discussion, when confidence limits are placed on the main correlation findings. To be sure the correlation coefficients are large, but how confident are we, given a single 11-year wavelet and with ENSO also involved dead center?

We changed the analysis, show also the non-normalized data, use p value (display only results for p<0.05). It is also shown that the significant correlation is not found in the main chimneys (out of most of the tropical rain forests).

2)  Pinto et al 2013

In this reviewer's opinion, the strongest single published result on lightning response (via thunder days) to the solar cycle was Pinto et al 2013. Why? Because multiple 11-year solar cycles were examined and because half a dozen recording stations in Brazil were analyzed, and finally because the details of the correlation calculations were shown, unlike earlier works by Brooks (1934) and Kleymenova (2006). The main problem here, not addressed by the authors, is that contrary to the inferred results in Figure 2a, an anti-phase behavior was found, with greater numbers of thunder days at solar minimum. The region of Brazil analyzed by Pinto et al. (2013) lies within the region of strong positive correlation coefficients in Figure 2a. Aren't these apparently contradictory findings troubling? If the authors are of the view that stroke counts make a dramatic improvement in the analysis, they need to show the evidence for that.

We now compare our results with previous studies. We found significant (p<0.05) in the region of South Atlantic anomaly, but not in the Amazon basin. Yes, our results partially contradict those obtained by Pinto et al. (2013). They were obtained in a different time interval as we discuss.

3)  The selected analysis period 2009-2022

Only a single solar cycle is represented by the 13-year period of WWLLN data available in this study. The fact that the period begins near solar min and ends near solar min, but is centered on the largest ENSO event if this century is not mentioned. The period 2014/15 was a Super El Nino event (Williams et al., 2021). Yes, the authors have taken averaging steps to suppress annual and ENSO cycles (lines 118 to 120) but nowhere do the authors suggest a possible aliasing by ENSO for the selected time interval. Enhance lightning in the phase of El Nino (and in the transition from cold to warm phase) is now well substantiated.

The possible influence of ENSO is now discussed. We also present separately the results without and with inter-annual filtering of lightning activity.

4)  The WWLLN data record

The authors have elected to do their solar cycle analysis on a lightning data record that has definite shortcomings. The authors are aware of the non-uniformity of the data (in space and in time) but are not sharing the quantitative details with readers. It would be valuable to see the history of number of available recording stations and the global stroke rates over the full 2009-2022 time interval. If trends are removed with quadratic fits, and that procedure leads to enhanced correlation coefficients, then we should be able to see how the data are treated as that impacts the findings. Traditionally, the Asian region is best

represented with WWLLN because the network developed out of R. Dowden in New Zealand, and Africa is least well represented because sensors there are fewer in number (mostly due to high internet costs).  Figure 1 does provide evidence for 3-chimney lightning dominance, but not with a ranking that is consistent with present knowledge (with Asia/Maritime Continent typically in third place).

We discuss the non-uniformity of data and use now the official correction coefficients that should deal with it at least partially. The controversial detrending using quadratic fits is no longer used.

5)  Reference to 'atmospheric electricity'

The controversial solar cycle- lightning issue has been around for a long time, and has been addressed in multiple publications involving the global electrical circuit that are neither cited nor critically addressed.  Lightning remains today a legitimate source term in the DC global circuit, and a solar influence has been considered more than once.   For example, Markson (1978) found evidence for solar modulation of the global circuit (though J. Willett had reservations).  Markson and Muir (1980) found evidence for solar wind influence on the global circuit.  Still later, Markson (1981) found evidence for positive correlation between ionospheric potential and cosmic rays.  The present study contends that cosmic rays are not influencing lightning intensity (and by inference, the DC global circuit).  Muhleisen (1977, follow on to Garmisch ICAE), in :"The global circuit and its parameters" found evidence for a solar cycle modulation of ionospheric potential.  The authors should be addressing these earlier findings in light of the results found in the present work.  In general, where the authors refer to "changes in the global electric circuit" (line 67), they need to expand the discussion.

We expended the discussion and reference the works by Markson at several places in the revised version.

6) Quantification of correlation coefficients in the text.

The correlation coefficient is the primary metric in this paper to characterize the solar cycle impact on lightning.  As such, more effort should be devoted to quantifying the numbers in the text wherever they are mentioned.   Specific examples of such places are lines 12, 14, 27-28, 40, 57-58, 169, 175, 193, 229, 232-234, 251, 257, 270-271 and 289.  Other works for which correlation coefficients are needed for comparison are Schlegel, Brooks and Ansol.

We added specific values of cross-correlation coefficients at convenient places and display now only statistically significant values ($p < 0.05$).

7)  Earlier thunder day analyses

In addition to the earlier study by Brooks (1934), another study by Kleymenova (1967), cited by Pinto et al 2013, is overlooked.  Brooks is here characterized as "thorough", and in that respect this study follows up on the classic investigation of thunder days (Brooks, 1925) which is indeed thorough, but in 1934 falls short in not showing actual data used in the correlation analysis.  Brooks (1934) also found exceptional correlation in Siberia, which is not followed up here.  Kleymenova (2006) found a mixture of phase in her solar cycle

studies, and phase clearly deserves more attention here.  Most importantly, if the present authors have evidence that stroke counts (with WWLLN) are improving on thunder days for this kind of analysis, that should be elaborated.  At the same time, the value of the thunder day observation should be critically discussed, given a consistent practice by meteorological observers since the late 19ᵗʰ century.  Modern lightning network data is limited in addressing climate issues on long time scales (like the present one).

We provide more references and comparison with previous work and included the analysis with simulated thunder days.

8) Temperature variations on the 11-year solar cycle

The global temperature is larger at solar max than at solar min, on account of the greater total energy received by the Sun at solar max.  The global temperature variation (of the order of 0.1 C peak-to-peak over 11 years) has been documented (Camp and Tung, 2007; Zhou and Tung, 2-13).  Nickolaenko (2015) had earlier suggested that a solar cycle in lightning activity and Schumann resonance intensity could be explained in this way.  Williams (2015) raised objections, but at least this in-phase relationship is consistent with the present findings and not consistent with Pinto et al. 2013, for example.  It is difficult for this reviewer to see how this physics is going to help explain positive correlation in only two of three lightning chimneys.

After recalculating the maps using corrected lightning data and displaying only regions with significant cross-correlation, the in phase relation is found east Africa, south-east part of South America, including part of South Atlantic, and west coast of Australia and part of Indian Ocean, but not in the typical wet rain forest areas, the Amazon basin in America, west part of Congo basin in Africa (in the lightning chimneys)

9) Solar ultraviolet radiation

The leading explanation for a positive phase relationship here, and one in keeping with item (8) above, involves heating by solar UV.  Surely some numerical models have been run to treat this additional energy source, and it would be helpful to the paper if the authors could chase the linkage from the heated stratosphere to the existence of enhanced upward motion in the troposphere, recognized as necessary to enhance the global lightning activity.  More attention needs to be given to connecting the UV with the air motions that influence lightning.

We changed the discussion

10) Planetary wave mechanisms

If the global lightning is the be enhanced by any kind of planetary wave activity.  Of relevance here are Anyamba et al JAS, and works on the Madden Julian Oscillation by U.S. authors Rutledge and coauthors and by Russian authors N. Slyunyaev, E. Mareev, Kozlova then it would be helpful to cite several works in which lightning is modulated by planetary waves and others. Additional info can be found in a review chapter on Schumann resonances in a book by Hans Betz in 2009.

We now discuss the Madden Julian Oscillation and provide some references (see Discussion)

11) Heliosphere Magnetic Field impact

Four pages of this 13-page paper are devoted to correlation checks on this mechanism. Given the space devoted, some additional explanation of the physical mechanism(s) suggested beyond what is stated in lines 64-67 on page 3 and at the bottom of page 8. "Changes in atmospheric electricity" is not sufficient here. Brian Tinsley is invested in a chain of events (including cloud microphysics) in this context but many of these links have yet to be verified.

We changed the text

12) Work by G. Satori on solar cycle time scale

Satori et al (2005) showed evidence for modification of the Schumann resonances on the 11-year time scale by virtue of the dramatic changes in solar X-radiation on the upper characteristic (i.e., magnetic) height of the Schumann cavity. These changes were shown to affect the modal frequencies and Q factors, but with smaller effects on intensities. Later Bozoki et al. (2021) demonstrated changes in Q factors by both changes in X-radiation and energetic electron precipitation, with corresponding increases in magnetic intensity that they linked with the Q factor increases. No changes in the lightning source on the 11-year time scale were inferred. The Kulak group in Poland also has a publication showing effects of solar X-radiation on cavity Q-factors on short time scales. This is a research area presently in a state of flux, with a recent article on SR measurements in China in JGR (Han et al., 2023).

Sátori (2011) also showed that the lightning area at high NH latitudes increases around solar maximum while it exhibits opposite behavior in the tropical/subtropical belts. Additional references are included later in this review.

We now cite the works by Satori and Bozoki and discuss possible biases of the detection efficiency due to changes in the Earth-ionosphere cavity.

13) Superbolt maximum

This pertains to the discussion in lines 264-269. I have been intrigued along with the third author about this important issue. How can changes in the ionospheric medium so strongly change the detectability of superbolts? Are we expected to have another superbolt maximum as solar maximum comes on again? Is the buildup to the 2014 El Nino responsible? One small detail here: the authors choose to remove seasonality in their correlation analysis, but this superbolt issue is definitely a seasonal issue. So what data are they pointing to in this work from boreal winter to substantiate this link?

We modified the text and discuss that winters should not have substantial influence on the results.

**Detailed Comments/Edits on the text:**

**Page 1**

Abstract:

line 12  What are typical numbers?

We modified the text as we now display only statistically significant coefficients (<0.05). Some numbers of the cross-correlation coefficients (up to about 0.9) are given in the Results section.

lines 20-21  Note possible inconsistency with Markson (1981)

We noted possible inconsistency with previous work and rephrased the sentence, but we avoid a citation in the abstract. The works by Markson are cited and discussed in the text at several places.

**Page 2**

Introduction

line 26  "thorough" is debatable.  See item (7) above.  Brooks also deserves more discussion.

We removed "thorough" and specified some numbers given by Brooks.

lines 34-35   The marked inconsistency between this work and Pinto et al 2013 is not noted and discussed.

We specified that Pinto mostly found anti-phase relation between SSN and thunder days. The works are compared in Discussion.

line 35  "thunder day data"

Corrected

lines 38-40   Were Schlegel's findings consistent or inconsistent with the present results, for the regions he investigated?

We now discuss it in Discussion section

line 40  What was a "significant cross correlation" by the authors' reckoning?

We changed the analysis (display for p<0.05)

line 41-44  Why did the authors not come back to this point in the Discussion?

We now provide results for thunder days too (simulated).

line 49  The authors are assuming here that correlation is causality, without further discussion.

We reformulated this sentence

lines 51-52  Need to get discussion of Markson (1981) in here somehow.

We added that Markson (1981) showed positive correlation between the ionospheric potential (atmospheric electric field) and cosmic rays and mentioned potential role of CR on cloud electrification

**Page 3**

lines 61-62   We need much more elaboration here, to see how the stratospheric warming will increase the vertical motions in the troposphere, known to affect lightning activity.

We reformulated this part, we refer to some previous studies. The exact mechanisms of stratospheric-tropospheric coupling still need to be investigated.

line 67   Elaboration on mechanism would also be useful here.

We specified the hypothesis given by Lam and Tinsley (2016) and pointed out that it needs to be verified.

line 72   See discussion of solar proton events in Markson (1978)

We expanded the discussion and added other references

lines 82-83  Need references for evidence here.  None of the suggested mechanisms link with CAPE and atmospheric instability, known ingredients for lightning activity,

We partly reformulate/specified the hypothesis suggested by Prykril et al. (2018). We agree that their hypothesis is very general and needs to be elaborated in more details.

**Page 4**

line 97 "data are obtained"

Corrected

Lines 97-104  The authors give no justification here for the selection of the WWLLN lightning data set to undertake this study.  A 30% detection efficiency for a 30 kA peak current threshold tells you right away that WWLLN is essentially an inefficient detector for CG lightning and with little access to the more dominant IC lightning.

We now justify/explain the reasons for the WWLLN selection and provide a brief comparison with LIS OTD detector here and in the beginning of the results Section.

line 108  More info is needed here about trends, and especially because a major ENSO event is occurring in the middle of the selected data window.

We reference the work by Holzworth et al. (2021) showing that the detection efficiency of WWLLN decreased before ~2013, causing the trend in data. In addition, we now show an example of raw WWLLN data in Section 3.

Line 115   If normalized variables are used then a short table is needed that shows the mean values and the standard deviations.

We now also present an example of raw data (Figure 3). Mean values and standard deviations are given in the related text.

line 118  The authors are aware of ENSO, but not facing up to its central presence in the selected data window.  More critical assessment is needed here.

Now discussed (see text after Eq 3) and Discussion.

line 119  "cross correlation coefficients"   You need to include the equations you use to produce the results.

Equation added (now Eq. 2)

**Page 5**

line 123  Yes the 3-year running mean will reduce the impact of ENSO will bill not eliminate it.  Too little discussion of "preparation" of the data is given here.

Now discussed (see text after Eq 3)

line 145  Why is this so?  Elaborate for non-experts.  What did you do with it?

Now discussed/explained (last paragraph of Section 2)

line 152 change "can be" to "are readily verified"

Done

line 153  Add a sentence:  "The continental lightning dominates the oceanic lightning by more than an order of magnitude".

Done

**Page 6**

line 160  Suggest repeating the entire period here.

Done

line 161   These numbers should be compared with state-of-the art results from LIS and GLD360 for numbers of strokes one gets with detection systems with much larger DE.

Yes, we agree. We reformulated, emphasizing that the given numbers concern the detected lightning, not the actual number of lightning, and briefly compare the WWLLN performance with the LIS OTD dataset (see also the preceding paragraph).

line 164 In light of apparently contradictory Pinto et al (2013) finding, the discussion on correlation phase needs to be expanded.

We modified the analysis and now present only cross-correlation regions in which the SSN is significantly ($p<0.05$) correlated and anti-correlated with lightning. In addition, we included an analysis of estimated number of thunder days for each bin. The results are compared with Pinto et al (2013) in Discussion section.

line 174   The text should tell what continent the selected bin is located and better yet, show the grid point on one of your maps.   It would also be helpful to see a selection of samples from bins not exhibiting maximum correlation.

We now marked with asterisks in Figure 3. An example with poor correlation is also presented

**Page 7**

Figure 2 caption, add "number of lightning strokes…"

Added

lines 186-187  "and atmospheric electricity" reference shows disinterest in details here, and references

We modified and extend this sentence.

line 190  No mention is made of the South Atlantic Anomaly region, often addressed in studies of this kind.

We now mention/discuss the South Atlantic Anomaly. It is now also mentioned in the abstract

**Page 8**

Figure 3 caption should say where this region is located and why it was selected.

Specified (now Figure 4 and 5), also in the text and marked in Figure 3.

Figure 3: clearly more lightning at solar maximum, and not what Pinto et al (2013) with thunder day analysis.

Yes, discussed in the Discussion section

Line 201 "changes of atmospheric electricity" Here again it sounds like atmospheric electricity is some nebulous subject not being addressed by the authors. We need to hear the details here.

As the results based on $|By| > 3$ nT did not bring much interesting or new. In addition, the focus of the paper is not on polar region where the lightning is rare, we removed this part.

**Page 10**

line 229 Quantify "large values"

Specified

**Page 11**

line 237 Here the authors seem concerned with phase. GOOD.

Thank you

**Page 12**

line 251 Quantify the values

as described we now display only regions with statistically significant ($p<0.05$) correlation.

lines 255-257 This sounds hand-wavy and unreferenced. It would be valuable to look at the literature and see where planetary wave activity (i.e., MJO) is affecting lightning activity.

Discussion was significantly changed and is more specific now, the MJO is mentioned and references added

**Page 13**

Line 258 Here the authors are attentive to phase.

Thank you

line 259 The monsoon trough is not a productive region for thunderstorms (see Williams et al., JAS, 1991), and it is not clear that a different mechanism is operating in the so-called break period between monsoon trough visitations. This is an unconvincing explanation for the different correlation behavior in the third chimney.

We removed the sentence related to monsoon.

Line 263  Early in the paper the authors indicated they were removing seasonal behavior by appropriate filtering.  What then are they pointing to to show results for boreal winter?

Using the corrected lightning instead of questionable removal of trends by quadratic function, we have not observed significant correlation of lightning with SSN in Europe, so we partly removed it.

lines 270 to 272   What is the physical basis here?

We modified the text as the results partly changed, using corrected counts. The focus is now on South Atlantic anomaly

line 274 to 277  This is quite hand-wavy.  Needs further substantiation.

We significantly modified this part

lines 278-280  The authors need to square their findings with Markson (1981).

We compare/discuss our results with Markson (1981)

Line 279  I do not understand the point here about "suitable weather conditions"

Explained (conditions leading to thunderstorm development) and modified.

lines 284 to 286  Why is this the case?  The authors need to greatly beef up the comparative results using thunder days and using measured strokes.

We modified the text.

line 286 "a solar rotation period signal in lightning occurrence"?  Please clarify.  See also Satori references below.  See also Anyamba et al  (2000) reference.

Modified and clarified.

line 291   The authors need to be more specific about the diverse mechanisms for the occurrence of thunderstorms and how that relates to the present study.

Removed and modified

**Page 14**

Line 294  Yes, analysis of a single solar cycle has definite limitations.

We agree

**References to add and discuss in this paper:**

We added most of the references and some others

Anyamba et al JAS 2000

Beloglasev and Akhmetov (2010)  Geomagnetism and Aeronomy

Girish and Eapon JASTP 2008.

Kleymenova (2006)  (citation is in Pinto et al. 2013)

Kozlova, A.V., N.N. Slyunyaeva, N.V. Ilina, F. G. Sarafanova, and A. V. Frank-Kamenetsky, The effect of the Madden–Julian Oscillation on the global electric circuit, Atmospheric Research, (**in press**), Dec 2022.

Markson (1978)

Markson and Muir (1980)

Markson (1981)

Muhleisen (1977)

Nickolaenko, A.   Sun and Geosphere, (2015).

Satori, G. doctoral thesis work (2011) : real-d.mtak.hu/512/

and on the solar rotation period see: Sátori, G ; Zieger, B

Areal Variations of the Worldwide Thunderstorm Activity on Different Time Scales as Shown by Schumann Resonances

In: Serge, Chauzy; Pierre, Laroche (szerk.) Proceeding of the 12th ICAE, Global Lightning and Climate

(2003) pp. 1-4. , 4 p.

1.  Willett comments on Markson (1978)

Williams, E., Sun and Geosphere, (2015)

End review

Earle Williams

November 26, 2023

**Reviewer 2**

The paper by Chum et al. presents the results of a simple correlation study between lightning activity (LA) and several solar/heliospheric indices and the corresponding geographical patterns. The study covers 14 years (2009-2022) and is focused on the 11-year solar cycles. While the finding of a strong correlation between the sunspot number (SSN) and LA in the African and Latin American regions is interesting, the paper is not recommended for publication in its present form because of essential methodological and conceptual flaws as detailed below.

*Thank you for your careful reading of the manuscript and for your useful comments. We believe they have contributed to the improvement of the manuscript.*

*General response. We carefully considered the comments raised by both reviewers and significantly changed the analysis and rewrote substantial part of the manuscript. The main changes are:*

- *The analysis is now based on the corrected WWLLN counts using the official correction coefficients to account for changes in detection efficiency (instead of previously used questionable detrending using quadratic function).*
- *A comparison with thunder days (simulated) used in previous studies is also provided.*
- *Only areas with statistically (95%) significant correlation (anti-correlation) are now shown in the maps, i.e. for p<0.05.*
- *The changes led to partially (not completely) different results. Accordingly, the text and discussion was changed. For example, the significant correlation in the region of South Atlantic anomaly is more clear. Also, it can be seen that most of the tropical rain forests are not significantly correlated.*

- Data used for the analysis of solar/heliospheric indices must be shown. SSN and NM time profiles are shown in Fig.3 but By, Bz and E_KL are not shown. The annually averaged By and Bz data in the OMNIWeb database vary between [-0.5,0.6] and [-0.3, 0.3] nT, respectively, never exceeding the values of 3 nT and 1 nT, as discussed in Figures 5 and 6. The authors must explain what the datasets they used are. Also, the authors need to specify in what coordinate system By and Bz are defined – GSE or GSM.

*The components By and Bz are in GSE coordinate system. Now specified in Section 2.*

*We now do not show the analysis for |By| > 3 nT as it did not bring much interesting or new. In addition, the focus of the paper is not on polar region where the lightning is rare, we removed this part.*

*(To explain, yearly means of were computed only from intervals when $|B_y| > 3$ nT.)*

- The methodology is unacceptably flawed.

  1. The authors study the cross-correlation coefficient, which is not defined but supposedly it is the linear Pearson's correlation. However, they do not estimate the statistical significance of the correlation which is crucially important for the short data series (14 data points for the full series and even less for the data shown in Figs. 5 and 6) and especially for the smoothed data series. The value of the cross-correlation coefficient alone is not informative – its statistical significance must be evaluated.

     *The equation for cross-correlation coefficient is shown now (Eq. 2). We now evaluate the statistical significance and display only results with significance larger than 95% (p<0.05), The results for smoothed and unsmoothed data are shown separately.*

  2. It is not clear what annually averaged values of and are. They both vary around zero not representing the IMF value. For , the annual values are mostly defined by the interplay between towards/award sectors and solar wind speed (By ~ v*B), while by heliospheric disturbances including CMEs.

     *Yes, the annual averages of $B_y$, $B_z$ are small. Therefore, we also present results for the Kan-Lee electric field (though they are similar as for $B_y$) because it is proportional to the transverse component of HMF ($\sqrt{(B_y^2 + B_z^2)}$).*

  3. The use of smoothed datasets distorts the correlation analysis and must be carefully evaluated.

     *The results for smoothed and unsmoothed data are shown separately to see the differences*

  4. The use of detrending (line 110+) is not substantiated nor properly analyzed.

     *The controversial detrending using quadratic fits is no longer used. Instead official correction coefficients were applied*

  5. 14 years is short to speak of the details of the 11-yr cycle (~1.3 full cycles).

     *We agree, we discuss the limitations*

- The logical chain is unclear. While the correlation analysis of SSN vs LA makes sense with Figure 2a being indeed interesting and worth discussing, the study of other indices is not motivated. For example, Fig. 2b is an obvious inversion of Fig. 2a because of the high anti-correlation between SSN and cosmic rays. As for other analyzed indices, the single pairwise correlation does not say anything about causal relations. The mechanisms are not properly discussed. I would expect to read about the ionospheric potential. Because of the dominant 11-year cycle in all indices, the relation between, e.g., By and LA can be via SSN which affects both. This can be formally tested with partial correlations, but this is not done.

  *We removed Figure 2b and significantly changed the text. The ionospheric potential is discussed (references to works by Markson, Kozlov).*

*We now also provide table of cross-correlation coefficients between the SNN and components of the HMF and Kan-Lee electric field.*

Some other minor comments:

- Equation (1) describe not normalisation but standardisation.

  *Done*

- Line 134: |Bz| = <|Bz|> or || ?

  *Modified, this part was removed*

- Line 144: the angle between By and Bz is 90 deg by definition.

  *Thank for noting, we corrected „clock angle of the transverse HMF (relative to the z-axis)"*

- Line 186: what do the authors mean by "the HMF polarity" – toward/away sectors or large-scale (dipole) polarity?

  *We modified. Yes, the cited papers dealt with the toward/away sectors of HMF defined by polarity (sign) of By (Bx has mostly the opposite polarity due to Parker spiral).*

- Line 260: How can sunspots "modify large scale modes of variability and atmospheric blocking"?

  *11-year solar cycle (represented by the Sun spot number – SSN) has been previously found to modulate frequency, persistence and location of Northern Hemispheric blocking. The exact mechanisms are still not well understood, possible processes include stratospheric responses to small changes in solar radiation through UV absorption and subsequent extratropical downward propagation of this signal.*

  *We have modified the text and added an additional reference for better clarity.*

---

## Editor Decision (ED1)

**Second review of: "Solar cycle signatures in lightning activity" by J. Chum, R. Langer, I. Kolmasova. O. Lhotka, J. Rusz and I. Strharsky**

The authors have done a thorough job in addressing the detailed comments and suggestions by the reviewers, and in comparing their findings with other results in the literature, and on this basis, the manuscript is much improved.  The statistics of the problem have also been much more thoroughly addressed.  But given that the results are so markedly different than in the first round, it would be valuable (at least for the reviewers' benefit) to know why the results are so different, and especially how confident the authors are in the end that there is a physical connection between the solar cycle and global lightning activity.

**Summary:  Consider for publication after minor revisions**

**Text edits:  (Note that line numbering is based on the "tracked changes" version of the revision.)**

Line 43 "around zero" is not very quantitative

Line 67  "e.g., Markson (1981)"

Line 88 What exactly was questioned by Hale (1979)?  This sentence is not clear.

Line 115 "using the World Wide Lightning Location Network"

Line 128 "that the satellite"

Line 131 "consistent" in what respect?  The sentence is unclear.

Line 135-136  These will generally be CG strokes so the 30% estimate for all lightning is misleading and inflated.  The WWLLN operators have never been straightforward about their detection efficiency estimates.  We know what the mean CG peak current is in lightning, so why not give a detection efficiency for that particular value of peak current, instead of a substantially larger one (30 kA)?

Line 143 "sensors"

Line 148  Yes, WWLLN has been weak in Africa because few sensors are located there.  The authors should also consult the paper by Virts et al. in BAMS which is perhaps the best paper to date on WWLLN performance in comparison with other optical detection systems.

Line 177 "logistic function"  What is it?  Why logisitic?

Line 207 "strokes"

Lines 212-213  You should have commented on this important aspect earlier in the manuscript.

Line 217  "also be shown"

Line 231 "the South Atlantic Anomaly region"

Line 236 "also shows up over a part of …"

Line 325 "number of lightning strokes"

Line 420  What does the MJO have to do with the main goal of the study, which is the 11 year solar cycle?

Lines 421-422  Positive variation?  What does this mean when the MJO is a global wave and Vi is a DC phenomenon?  Please clarify.

Line 422  "depend"

Line 423 "the ENSO"

Line 424 "occurs"

Line 425 "Schumann"

Line 427 change "solar" to "a"

Line 428 These events were not Super El Ninos (ONI index > 2 C) so how "reasonable" is this conclusion has not been clarified by the authors. Instead, they seems to be hoping that ENSO aliasing is not a problem. (I am reviewing another paper on lightning trends in the South China Sea, and ENSO aliasing has been a problem in establishing a decadal trend.)

Line 442 This text line (as are many others in my copy) is gibberish.

Line 443 "found a solar cycle"

Lines 445-446  Change "In contrary" to "In contrast"

Line 448 "over a non-negligible part"

Line 452-453  Do you mean to say "not uncorrelated"? (double negative).  This sentence is not as intended and needs to be rewritten.  It is not clear at present where the authors stand on Markson's (1981) claim that cosmic rays and ionospheric potential are anti-correlated.

Line 461  "the By component"

Line 465 "the South Atlantic anomaly"

Line 467 "Earth's"

Line 469 "a large flux"

Line 476 "The energy spectrum…"

Line 483 Gibberish again (unintelligible sentence, needs rewriting)

Line 488 Sentence remains unfinished.

Lines 496-497  Another incomplete sentence

Line 501  Sentence is difficult to read/decipher.

Line 503  Sentence is incomplete.

I don't see a real conclusion to this work (though the last sentence is covered up in my copy and may be very important).  Are the authors happy and content with their results?  It seems a collection of results

many of which bear little relation to other earlier findings. One of these is Pinto et al. (2013). Another is the strong Siberia result in Brooks (1934), as the present correlations in that region are mostly negative (anti-correlations).  The robust nature of positive correlations almost everywhere evident in the first round has largely disappeared.  The possible connection with the South Atlantic Anomaly is interesting.  What single additional effort do the authors view as being able to shed important new light on this problem?

End review

Earle Williams

April 14 2024

---

## Author Response (AR2)

*Reviewer #2*

*The authors have somewhat improved the manuscript but it is still far from possible acceptance. The level of statistical thoroughness is insufficient to judge the significance of the results and raises several major concerns. Accordingly, before this is clarified, the reliability of the claimed results cannot be assessed.*

**We thank the reviewer for the careful reading of the manuscript and thoughtful and improving comments.**

*1. Datasets. The authors ignored my previous requirement to show the annual Bz and By series which are not even described in how they were obtained. All the used datasets, including By, Bz, B||, BT, E must be shown and described. Without that, the results cannot be trusted.*

**We are sorry, we misunderstood your previous comment and thought you were interested in cross-correlation coefficients between the SSN and these quantities, which we put in Table 1 in the first revision.**

**Now, in the new revision, we show the time series of *By, Bz, $E_{||}$, and $E_T$* used for the construction of cross-correlation maps with lightning (Figure 5 and related text – lines 261-262 and 419-420). In addition, we also show the ENSO index.**

*2. The authors do not describe how they estimate the significance/confidence of the obtained correlation coefficients. This is particularly crucial for the smoothed data series since the significance/confidence of correlation between smoothed series cannot be estimated using the standard formula. The authors should explicitly describe how they evaluated the 95% conf. level for the smoothed series. I suspect that was done incorrectly leading to the overestimated results.*

**Thank you for this comment. Considering also the concern of the other reviewer on the effect of ENSO, we now provide cross-correlation maps with the ENSO index (Figure 10 and the related text – last paragraph of the Section 3) instead of smoothing lightning data in time. We completely removed the cross-correlation maps with the smoothed lightning data in the revised version**

**The statistical significance of correlation between data series (now only unsmoothed) is calculated by the corrcoef function in MATLAB. It is based on the use of the t-statistics with N-2 degrees of freedom. This information is added in the text (Section 2, text after Eq. 2).**

*3. Some correlation maps do not look statistically significant. For example, the fraction of the shown coloured bins in Fig.2a is ~5% which is fully consistent with the applied significance threshold (<0.05), viz. 5% of the correlation values may appear seemingly significant just by random coincidence. This should be clarified.*

**We now discuss a possibility of random coincidence in the end of the first paragraph of the last Section (Discussion and Conclusions). We also note that random coincidence is probable for the cross-correlation with Bz component.**

*4. In addition, there is a problem with the units of the analyzed data. This doesn't affect the results based on the standardized series but is quite annoying. For example, the authors discuss the number of all lightning strikes during the analyzed period (Fig.1) with the maximum values of ~3000 strikes (per 1x1 deg bin, as specified in line 208). However, this leads to 0.02 lightning/km2/yr which is obviously too small. Is it the number of strokes given per km2 or 1x1 deg bin? Fig.4c gives the values of 3E5 light strokes per bin which is inconsistent with Fig.1. Another example is Fig.4b which is declared as "Yearly NM counts" with the value of ~3E4 counts/yr which leads to an obviously wrong count rate of ~3 counts/hr. All the units must be verified.*

**Thank you for noting. Regarding Figure 1, we accidently plotted map of simulated thunder day numbers. We now provide both map containing corrected number of lightning (Fig 1a) and the number of simulated thunder days (Figure 1b).**

**Concerning Figure 4b, we corrected the description/caption. It is correctly "Yearly averages of 1-min NM counts" and the related text on lines 244-245.**

*Other minor comments include:*
*• Please write "sunspot number" (not "Sun spot number")*

**Corrected**

*• Line 168-169: what does "have been studied primarily for simplicity" mean?*

**To make it clear, we reformulated to "…**but since the relative changes in $B_y$, $B_z$ are much larger than the relative changes in the Earthward solar wind speed $v_x$, only the dependencies of the Earth's electric field on $B_y$ or $B_z$ have been often studied (e.g., Burns et al., 2008)**"**
* * *
**Reviewer #1 (E. Williams)**

*The authors have done a thorough job in addressing the detailed comments and sugges5ons by the reviewers, and in comparing their findings with other results in the literature, and on this basis, the manuscript is much improved. The sta5s5cs of the problem have also been much more thoroughly addressed. But given that the results are so markedly different than in the first round, it would be valuable (at least for the reviewers' benefit) to know why the results are so different, and especially how confident the authors are in the end that there is a physical connec5on between the solar cycle and global lightning actvity.*
*Summary: Consider for publicaton after minor revisions*
*Text edits: (Note that line numbering is based on the "tracked changes" version of the revision.)*

**We thank the reviewer for the careful reading of the manuscript and helpful comments.**

*Line 43 "around zero" is not very quantitative*

**Specified, "(absolute value less than 0.2)", on line 34**

*Line 67 "e.g., Markson (1981)"*

**corrected**

*Line 88 What exactly was questioned by Hale (1979)? This sentence is not clear.*

**reformulated "Hale (1979) suggested to look for effects more directly related to magnetospheric and auroral processes."**

*Line 115 "using the World Wide Lightning Location Network"*

**corrected**

*Line 128 "that the satellite"*

**corrected**

*Line 131 "consistent" in what respect? The sentence is unclear.*

**Reformulated, "The WWLLN lightning counts in 1°×1° bins are used in this study, but is also shown that similar results are obtained if larger bins (3° latitude × 6° longitude) are used."**

*Line 135-136 These will generally be CG strokes so the 30% estimate for all lightning is misleading and inflated. The WWLLN operators have never been straightforward about their detection efficiency estimates. We know what the mean CG peak current is in lightning, so why not give a detection efficiency for that particular value of peak current, instead of a substantially larger one (30 kA)?*

**Specified that the detection efficiency concerns the CG strokes (line 123)**

*Line 143 "sensors"*

**corrected**

*Line 148 Yes, WWLLN has been weak in Africa because few sensors are located there. The authors should also consult the paper by Virts et al. in BAMS which is perhaps the best paper to date on WWLLN performance in comparison with other optical detection systems.*

**Reference to Virts et al. added (lines 132-134)**

*Line 177 "logistic function" What is it? Why logistic?*

**Text around Eq (3) was reformulated/expanded. We use the logistic function to avoid a fixed threshold and to avoid a small uncertainty around it.**

*Line 207 "strokes"*

**corrected**

*Lines 212-213 You should have commented on this important aspect earlier in the manuscript.*

**Now mentioned also in section 2 (lines 136-137)**

*Line 217 "also be shown"*

**corrected**

*Line 231 "the South Atlantic Anomaly region"*

**corrected**

*Line 236 "also shows up over a part of …"*

**The whole text and Figures related to smoothed data were removed. See the response to comments of the reviewer 2 please.**

*Line 325 "number of lightning strokes"*

**corrected**

*Line 420 What does the MJO have to do with the main goal of the study, which is the 11 year solar cycle?*

**Reformulated. MJO can affect rain intensity and lightning and there was a suggestion in the literature that it is linked with the ionospheric potential (line 355-356).**

*Lines 421-422 Positive variation? What does this mean when the MJO is a global wave and Vi is a DC phenomenon? Please clarify.*

**Reformulated. MJO can affect rain intensity and lightning and there was a suggestion in the literature that it is linked with the ionospheric potential (line 355-356).**

**Reformulated.**

*Line 422 "depend"*

**corrected**

*Line 423 "the ENSO"*

**corrected**

*Line 424 "occurs"*

**corrected**

*Line 425 "Schumann"*

**corrected**

*Line 427 change "solar" to "a"*

**changed**

*Line 428 These events were not Super El Ninos (ONI index > 2 C) so how "reasonable" is this conclusion has not been clarified by the authors. Instead, they seems to be hoping that ENSO aliasing is not a problem. (I am reviewing another paper on lightning trends in the South China Sea, and ENSO aliasing has been a problem in establishing a decadal trend.)*

**We changed the text (lines 366-371 and also 362-263) . In addition, we now provide the maps of significant cross-correlation coefficients between the ENSO index and lightning counts (Figure 10a) and thunder days (Figure 10b).**

*Line 442 This text line (as are many others in my copy) is gibberish.*

**We are sorry it was a problem of pdf file (version) containing track changes. We now provide text with highlighted changes (yellow).**

*Line 443 "found a solar cycle"*

**corrected**

*Lines 445-446 Change "In contrary" to "In contrast"*

**corrected**

*Line 448 "over a non-negligible part"*

**The whole text and Figures related to smoothed data were removed. See the response to comments of the reviewer 2 please.**

*Line 452-453 Do you mean to say "not uncorrelated"? (double negative). This sentence is not as intended and needs to be rewritten. It is not clear at present where the authors stand on Markson's (1981) claim that cosmic rays and ionospheric potential are anti-correlated.*

**Reformulated. "cosmic rays are uncorrelated with lightning activity over most of the globe."**

*Line 461 "the By component"*

**corrected**

*Line 465 "the South Atlantic anomaly"*

**corrected**

*Line 467 "Earth's"*

**corrected**

*Line 469 "a large flux"*

**corrected**

*Line 476 "The energy spectrum…"*

**corrected**

Line 483 Gibberish again (unintelligible sentence, needs rewriting)

**We are sorry it was a problem of pdf file (version) containing track changes. We now provide text with highlighted changes (yellow).**

Line 488 Sentence remains unfinished.

**We are sorry it was a problem of pdf file (version) containing track changes. We now provide text with highlighted changes (yellow).**

Lines 496-497 Another incomplete sentence

**We are sorry it was a problem of pdf file (version) containing track changes. We now provide text with highlighted changes (yellow).**

Line 501 Sentence is difficult to read/decipher.

**We are sorry it was a problem of pdf file (version) containing track changes. We now provide text with highlighted changes (yellow).**

Line 503 Sentence is incomplete.

**We are sorry it was a problem of pdf file (version) containing track changes. We now provide text with highlighted changes (yellow).**

I don't see a real conclusion to this work (though the last sentence is covered up in my copy and may be very important). Are the authors happy and content with their results? It seems a collection of results many of which bear little relation to other earlier findings. One of these is Pinto et al. (2013). Another is the strong Siberia result in Brooks (1934), as the present correlations in that region are mostly negative

(anti-correlations). The robust nature of positive correlations almost everywhere evident in the first round has largely disappeared. The possible connection with the South Atlantic Anomaly is interesting. What single additional effort do the authors view as being able to shed important new light on this problem?

**The larger regions of positive correlation in the first submitted version were based on the trend removal using quadratic polynomial and on smoothing lightning data over neighboring years. Based on the critical comments made by the reviewers, we do not use these methods any more. We now only use official correction coefficients to the WWLLN data and in the second revision, we also do not use smoothing of the lightning data. Instead of that cross-correlation with the ENSO index is provided.**

**Yes, we agree that our study does not confirm some of the previous works. Note that it is done over different time intervals and that also the results in the literature differ (see also the Discussion section). We agree that the most interesting results is the connection with the South Atlantic Anomaly. It does not depend whether lightning counts or thunder days are used. As we discuss, further investigations are needed.**